# Implementing Model Predictive Control and Steady-State Dynamics for Lane Detection for Automated Vehicles in a Variety of Occlusion in Clothoid-Form Roads

**DOI:** 10.3390/s23084085

**Published:** 2023-04-18

**Authors:** Swapnil Waykole, Nirajan Shiwakoti, Peter Stasinopoulos

**Affiliations:** School of Engineering, RMIT University, Melbourne, VIC 3000, Australia

**Keywords:** lane detection and tracking algorithm, advanced driver assistance system, intelligent vehicle, custom datasets, complex road geometry

## Abstract

Lane detection in driving situations is a critical module for advanced driver assistance systems (ADASs) and automated cars. Many advanced lane detection algorithms have been presented in recent years. However, most approaches rely on recognising the lane from a single or several images, which often results in poor performance when dealing with extreme scenarios such as intense shadow, severe mark degradation, severe vehicle occlusion, and so on. This paper proposes an integration of steady-state dynamic equations and Model Predictive Control-Preview Capability (MPC-PC) strategy to find key parameters of the lane detection algorithm for automated cars while driving on clothoid-form roads (structured and unstructured roads) to tackle issues such as the poor detection accuracy of lane identification and tracking in occlusion (e.g., rain) and different light conditions (e.g., night vs. daytime). First, the MPC preview capability plan is designed and applied in order to maintain the vehicle on the target lane. Second, as an input to the lane detection method, the key parameters such as yaw angle, sideslip, and steering angle are calculated using a steady-state dynamic and motion equations. The developed algorithm is tested with a primary (own dataset) and a secondary dataset (publicly available dataset) in a simulation environment. With our proposed approach, the mean detection accuracy varies from 98.7% to 99%, and the detection time ranges from 20 to 22 ms under various driving circumstances. Comparison of our proposed algorithm’s performance with other existing approaches shows that the proposed algorithm has good comprehensive recognition performance in the different dataset, thus indicating desirable accuracy and adaptability. The suggested approach will help advance intelligent-vehicle lane identification and tracking and help to increase intelligent-vehicle driving safety.

## 1. Introduction

With the fast development of high-precision optical sensors and electronic sensors, as well as high-efficiency and highly effective computer vision and machine learning algorithms, real-time driving scene comprehension has become more practical. Many academic and industrial research organisations have committed significant resources to the development of sophisticated algorithms for driving scene interpretation, with the goal of developing either an autonomous car or an advanced driver aid system (ADAS). Lane identification is one of the most fundamental study areas in driving scene interpretation. After obtaining lane locations, the car will know where to proceed and will avoid colliding with other lanes [1].

In recent years, a variety of lane detection algorithms with sophisticated performance have been presented and described in the literature. Among these approaches, some use geometry models to describe the lane structure [2,3], while others express lane identification as energy minimisation issues [4,5], and yet others segment the lane using supervised learning algorithms [6,7,8,9]. However, most of these approaches only provide solutions for detecting road lanes in the currently active frame of the driving scene, leading to poor performance in dealing with difficult driving scenarios, such as heavy shadows, severe road mark degradation, and serious vehicle occlusion. In certain cases, the lane may be anticipated in the wrong direction, only partially identified, or not detected at all. The information offered by the present frame is insufficient for effective lane recognition or prediction, which is one of the key reasons for the need for further developments in this area.

In general, road lanes are continuous line constructions on the road surface that appear either solid or dashed. The location of lanes in adjacent frames is closely connected because the driving scenarios are continuous and heavily overlapped between two surrounding frames. More specifically, the lane in the current frame may be predicted using many prior frames, even if the lane has been damaged or degraded due to shadows, stains, and occlusion. This inspires us to investigate lane recognition using photos of a continuous driving scenario.

Object detection [10,11,12], image classification/retrieval [13,14,15,16], and semantic segmentation [16,17,18,19] are just a few of the computer vision problems that deep learning has been shown to solve at or above human levels. Deep neural networks are classified into two categories. The first is the deep convolutional neural network (DCNN), which is an example of a neural network that performs well when it comes to feature abstraction for images and video by analysing the input signal using multiple layers of convolution. The other is the deep recurrent neural network (DRNN), which recursively analyses the input signal by breaking it down into subsequent blocks and constructing complete connection layers between them for status propagation and is skilled at information prediction for time-series signals. Considering the aforementioned difficulty in lane recognition, it would appear that the time series represented by the continuously captured images of the driving scene can be analysed by DRNN.

The goal of the previously mentioned trajectory tracking controllers is to reduce the lateral error between the autonomous vehicle and the reference trajectory. The issue develops for the majority of path-tracking controllers when the autonomous vehicle goes through a severe bend in the reference route [20]. Driving through a severe curve in the trajectory is always risky if the vehicle’s speed and steering angle are not properly regulated. In the actual world, a motorist must be able to lower their speed while still controlling the steering wheel in order to remain on the road and safely navigate the tight bend. When monitoring a reference trajectory, an autonomous vehicle should follow a similar concept.

This study proposes an innovative path tracking approach, the major contribution of which is to decrease the lateral errors and tracking errors of the autonomous vehicle, particularly while traversing steep bends, by constructing a hybrid longitudinal and lateral control system. For speed control, the longitudinal control design is based on geometrical curvature information and is combined with the Model Predictive Control (MPC). The intended speed of the vehicle is calculated using the vehicle’s current speed and the road’s curvature profile. Furthermore, the feedforward optimum preview control is utilised to address lateral errors in the trajectory by controlling the vehicle’s steering angle, which is caused by road curvatures. The preview capability is used to reduce the mistakes caused by external and environmental disturbances. The proposed algorithm’s tracking performance is measured using the average root-mean-square errors. Both simulation and experimental findings are used to evaluate and validate the efficacy of the proposed strategy. 

The paper’s main contributions are twofold. First, to address the issues involved in lane detection and the tracking algorithm, such as occlusion on a clothoid road, a solution based on MPC preview capability is designed and successfully deployed to prevent occlusion by controlling the lateral error on the reference path and increasing the lane detection performance. The existing models experience problems with occlusion in inclement weather conditions, leading to significant concern over passenger safety. Further, clothoid-shaped roads have been investigated far less in the existing literature. To address this issue, the current paper presents a model based on the differential equation of motion that will calculate the kinematic steering angles needed to keep the vehicle in the desired direction of motion. The developed model determines the actual direction of motion by considering vehicle’s sideslip when calculating the direction of motion. This is achieved by solving the vehicle’s dynamic equations for a desired direction of motion. A learning-based approach for lane detection utilising continuous driving scene images is presented to address the issue that a lane cannot be reliably recognised using a single image in the presence of shadow, road mark deterioration, and vehicle occlusion. Because more information can be derived from numerous continuous images than from a single current image, the suggested technique may forecast the lane more accurately, particularly when dealing with the aforementioned problematic conditions. 

Second, a mathematical model of the vehicle is developed for the purpose of designing the path tracking controllers in this work. We created the vehicle model while taking into account inertial co-ordinate dynamics. It has been demonstrated in earlier studies that building MPC controllers based on vehicle models of various complexity requires a lot of effort, and tuning is challenging due to more complicated vehicle models. A simplified “four-wheeler” model with a linear tyre model is chosen in this work because the goal of this research is also to determine how to track the appropriate trajectory quickly and steadily, which pertains to vehicle handling stability.

The structure of this paper is as follows. Section 2 presents the key parameters of the algorithm. Section 3 introduces the proposed mathematical model for the lane detection algorithm that includes MPC preview capability strategy and steady-state dynamics. Section 4 reports the performance of the proposed algorithm in clothoid-form roads under a variety of occlusion. Section 5 concludes our study and briefly discusses potential future work.

## 2. Key Parameters of Algorithm

In the next few sections, the key parameters of the mathematical model are obtained, and the MPC preview strategy is developed and implemented. Steady-state dynamic equations are used to calculate the steering angle, yaw angle, and sideslip of the vehicle in the desired path, and a mathematical model is developed and tested in a simulation environment on clothoid-shaped structured and unstructured roads. 

### 2.1. Linear Tyre Model

To build the vehicle dynamics model and design the control convolution stages, a tyre model must be created. The development of an accurate tyre model is critical for vehicle dynamics simulation and vehicle handling stability research. The linear tyre model proposes a straight-line relationship between lateral force and slip angle. Figure 1 shows how to calculate the lateral force of a tyre.

The general form of the magic formula is as follows:(1)$$Y\left(x\right)=Dsin\left(C\;arctan[Bx-E(Bx-\arctan\left(Bx\right))]\right)$$
where *Y* denotes the output variable, *x* denotes the input variable, *D* denotes the peak factor, *C* denotes the shape factor, B denotes the stiffness factor, and *E* denotes the curvature factor. Only the effects of the tyre sideslip angle, vertical load, and road adhesion coefficient on the cornering force are discussed in this article. The formula has a higher fitting accuracy and can still be used outside of the limit value, demonstrating good robustness.

### 2.2. Forces Acting on Passengers

The most frequent strategy utilised to improve passenger comfort is to optimise the vehicle movement to reduce forces and jerks. A proper seat and suspension design might reduce the vertical forces and vibrations caused by road disturbances. Horizontal forces result from steering and acceleration. Passengers’ vertical oscillations are highlighted in [20], and the researchers proposed that the ISO-2631-1 standard misunderstands passenger comfort parameters, including lateral oscillations for seated passengers. 

Smooth control is clearly preferred to avoid overshooting and minimise resulting forces. To assist tracking, continuous trajectories might be generated. Path continuity was noted in [21]. The intricacy of its synthesis and real-time execution prevented their employment in time-critical applications such as highway navigation. Clothoid usage was confined to parking assist devices, and parametric vector-valued curves with continuous curvature, velocity, and acceleration were suggested. It is simple to implement these planning approaches with trajectory tracking algorithms to reduce tracking errors and overshooting [22]. Planning, generating, and tracking paths are supposed to reduce load disruptions, and we already utilise acceptable longitudinal jerk and acceleration approaches for passenger comfort and safety. 

### 2.3. Designing a Control Strategy

Predicting the process output at a limited control horizon based on historical and present values is known as prediction. The optimiser calculates a control sequence based on the cost function and constraints and then repeats the procedure in a receding horizon. MPC outperforms conventional control strategies in trajectory tracking because it can manage both “soft” and “hard” constraints on state variables and control the input/output, which enhances performance and stability. As a result, the vehicle’s lateral stability and path-following accuracy may be improved (lane detection). Using the aforementioned technique, the front steering wheel angle is controlled by the MPC preview capability (PC).

The perpendicular distance to the path determined from the vehicle’s centre is required by the controller. This indicates that a straight line between the vehicle and the path is not always the case, since it might be a non-perpendicular line (Figure 2 illustrates this idea). From the vehicle to the path, the dotted red line is the shortest. The perpendicular line from the vehicle centre to the path represented by ye, the mistake in the lateral position, is shown by the red line in Figure 3. Two-line equations are used to answer this problem. A curve is fitted to a tiny section of the path and the other line is drawn using the vehicle’s heading angle. From the vehicle’s centre, a line is drawn with a slope equal to the heading angle +90°. With a fairly basic equation, y = ax + b, a line perpendicular to the vehicle is created.

By locating the intersection point of these two lines, the perpendicular point may be obtained. As a consequence, the point is ($p_{x}$, $p_{y}$). The difference between (*c_x_*, *c_y_*) and (*y_e_*) is now the distance *y_e_* ($p_{x}$, $p_{y}$). The perpendicular distance of the projected vehicle may be calculated using the same method. To achieve the point *c_x_*, *c_y_*, there is one more step: add the length L in the direction the vehicle is heading. In the future, both of these distances will be utilised in the system. *y_e_* will be used to assess performance and will be an input to the controller.

In Figure 4, t indicates the current time, X and Y represent the longitudinal and lateral position in the inertial co-ordinate system, respectively, and p and T represent the MPC prediction horizon and PC preview time, respectively. The MPC theory predicts state variables at each sample point within a prediction horizon. With the PC and the estimated state variable, the reference yaw rate is calculated at each sample point. In Figure 4, the dotted red line shows the PC reference route, whereas the black line indicates the MPC reference path. The red line outside the MPC prediction horizon shows the reference yaw rate, while the green line shows the reference lateral displacement. The reference yaw rate and lateral displacement are used to develop the MPC optimiser. Figure 4 shows a metonymical strategy to increase the original basis’s effective reference trajectory range. By expanding the effective reference path length, the vehicle’s lateral stability and path-following accuracy may be increased.

### 2.4. Implementation of Strategy

This section briefly presents the preview capability (PC). The car moves according to the Ackerman mechanism, where R and V are the radius of the road curve and the vehicle speed, respectively. In Figure 5, F(X) is the reference path equation and y = Y(*t*). The preview distance is d and the preview time is T = d/V and G is the rigid support. The steering angle is defined by the trajectory curvature 1/R and lateral acceleration Y(*t*). After T, the vehicle’s lateral displacement is stated as follows:(2)$$Y(t+T)=Y(t)+TY(t)+\frac{T^{2}}{2}Y(t)$$

The best trajectory curvature and lateral acceleration for aligning Y (t + T) with F (X (t + T)) are:(3)$$Y(t)=\frac{2}{T^{2}}[\mathrm{FX}(t+T)-Y(t)-\mathrm{TY}(T)]$$

In the vehicle steering moves, the yaw rate *j* is decided by the vehicle speed and the trajectory curvature:(4)$$\varphi =\frac{V}{R}$$

Figure 4 demonstrates the major components of a path-tracking system that uses MPC-PC to estimate the front wheel steering angle. The reference generation module estimates the vehicle condition and the reference yaw rate. The lateral acceleration measured by the PC changes the predicted vehicle condition at each sampling time. The reference yaw rate is produced by the consistent procedure of obtaining the ideal preview lateral acceleration and predicting vehicle states. To reduce the accumulating error and provide a more accurate reference yaw rate, rolling computation is required. The reference consists of two modules: lateral displacement and yaw rate. This is because the lateral displacement is determined by the reference path in the predictive horizon, whereas the reference yaw rate is determined by a desired trajectory in the preview distance. Unlike the conventional MPC, this approach expands the effective reference path without further computation. The model predictive controller (MPC) optimiser with input and output constraints calculates the steering wheel angle to follow the predefined path, and the solution becomes a quadratic programming problem with constraints.

### 2.5. Steady-State Dynamic Equation

We designed sample roads (straight and clothoid) and used them to determine the kinematic steering angle required to keep the vehicle in the desired path of motion. The next step is to identify the actual path of motion. This situation refers to when the sideslip of the vehicle is considered when determining the real path of motion and can be accomplished by solving the vehicle’s dynamic equation of motion for a desired path. The clothoid and straight-line equations were used to develop the sample road but, after two or three steps, they became complicated to solve analytically. After solving these equations for the sample road, it was confirmed that the equations of motion are impossible to solve in advance due to complications; however, the steady-state response can be used as an appropriate replacement. The aforementioned scenarios were investigated by solving the differential equations of motion and introducing a new method, i.e., steady-state dynamic control.

## 3. Design of the Mathematical Model

The four-wheeler model is constructed using Newton’s second law of motion, and the equations for lateral and yaw motion may be stated as follows:(5)$$ma_{y}=\sum F_{y}$$
(6)$$I_{z}r=\sum M_{z}$$

m = mass of the vehicle;*a_y_* = lateral vehicle acceleration.

The total of forces in the *y*-direction and the maximum moment of the *z*-axis are as follows:(7)$$\sum F_{y}=F_{yf}+\cos\delta +F_{yr}$$
(8)$$\sum M_{y}=lF_{yr}+\cos\delta -F_{yr}$$

The *y*-direction acceleration $a_{y}$ is composed of two components: the acceleration $v_{y}$ and the rotational effect $v_{x}$, where r is the angular velocity around the *z*-axis, commonly known as the yaw rate. Thus:(9)$$a_{y}=v_{y}+rv_{x}$$

### 3.1. Stability

The stability of the linear dynamic four-wheeler model is examined in this subsection. Since this is a linear system, stability can be established if the eigenvalues contain no positive real components. Vehicle instability is usually produced by a sudden tripping rollover or a non-tripping rollover. As a result, while examining such vehicles in path tracking, the problem of roll stability control must be taken into account.
$$\mathrm{LTR}\;(\mathrm{lateral}\;\mathrm{transfer}\;\mathrm{ratio})=\frac{\left(F_{front}-F_{rear}\right)}{\left(F_{front}+F_{rear}\right)}$$

The idea is to derive the lateral transfer ratio (LTR) index by feeding the vehicle’s real-time tyre vertical force to the controller, comparing it to the LTR threshold index, and determining whether there is a rollover hazard based on preview capability control theory. After taking into account the effect under high-speed and low-adhesion conditions, the LTR rollover threshold is adjusted to LTR = 1.

### 3.2. Longitudinal Dynamic

The objective of longitudinal control in this research is to ensure that the longitudinal speed of the vehicle’s centre of mass ($v_{x}$) is the same as the desired longitudinal speed (v). In order to use the sliding mode approach [23], it is necessary to determine a relationship between the longitudinal speed and the applied torques to the wheels.

However, this research suggests that the steering angle and lateral forces be used together in the prior step in order to increase accuracy. The next section is a short description of the equations used in calculating the control inputs.

The slip surface is evaluated according to the following equation:(10)$$s_{x}=(v_{x}-v_{R})$$

We may now differentiate the slip surface $S_{x}$ and suppose that it is equal to zero as a result. The equation is obtained by substituting the corresponding term from Equation (10) for $v_{x}$, resulting in the expression:(11)$$(F_{x}\cos\delta -F_{y}\sin\;\delta +F_{z})+v_{y}-v_{R}=0$$

The symbol that appears in the equation represents the steering angle of the previous step that is known. After simplifying the relationship, the longitudinal force for each tyre is replaced by its equivalent.
(12)$$T_{t}=R_{w}(f_{t}F_{z}\;\cos\delta +f_{x}F_{z}+f_{y}\;\sin\delta +F_{aro})-mv_{y}+\mathrm{mv}$$

The acceleration torque delivered to the front and rear wheels may be calculated using the second rule.
(13)$$T_{f}=T_{t}\;\mathrm{and}\;T_{\tau }=\frac{F_{x}}{F_{y}}T_{t}$$

### 3.3. Tyre Forces and Angle

At low slip levels, the longitudinal and lateral forces are mostly determined by the tyre’s elastic characteristics; however, as the slip angle grows, the contribution reduces and the friction between the tyres and the road increases gradually. 

The kinematic condition is applied for both steering angles. Equations (2)–(4) depict the vehicle’s kinematics in accordance with the geometry connection depicted.
$$ma_{y}=F_{f}sin\delta_{f}+F_{r}cos\delta_{r}+F_{r}$$
$$x_{0}=vcos(\varphi +\beta )$$
$$y_{0}=vsin(\varphi +\beta )$$
φ=ϵ
where longitudinal and lateral co-ordinates ($x_{0}$ and $y_{0}$) are located at the centre of gravity (CoG) and are the vehicle’s yaw angle, yaw rate, and sideslip angle, respectively. Assuming that the path’s curvature is minimal, it creates minor variations in the vehicle’s yaw angle and sideslip angle. As a result, the kinematic model may be depicted as:$$x_{0}=v$$
$$y_{0}=v(\varphi +\beta )$$
φ=ϵ

When the vehicle’s speed increases and the curvature of the road changes, it is difficult to monitor the trajectory using merely the kinematic model of the vehicle. If the vehicle’s longitudinal velocity is considered to remain constant, the dynamics of the vehicle may be shown using Newton’s law, as follows:$$a_{y}=v(\beta +\varphi )$$
where $a_{y}$ is the starting acceleration at the CoG in the *y*-axis direction. The acceleration *v* along the *y*-axis and the centripetal acceleration *v* both contribute to $a_{y}$. As a result, the vehicle’s lateral angular motion equation may be represented as:$$mv\left(\varphi +\beta \right)=F_{f}+F_{r}$$

The equation for yaw dynamics in the *z*-axis is as follows:$$I_{z}=I_{f}F_{f}-I_{r}F_{r}$$
where *I_f_* and $I_{r}$ are the distances of the front and rear tyres from the vehicle’s centre of gravity, respectively, $F_{f}$ and $F_{r}$ are the front and rear lateral tyre forces, and $F_{f}$ and $F_{r}$ are defined as:$$F_{f}=C_{f}\alpha_{f}$$
$$F_{r}=C_{r}\alpha_{r}$$
where $C_{f}$ and $C_{r}$ are the front and rear tyre cornering stiffnesses. The following are the front and rear tyre sideslip angles of vehicle *f* and *r*:$$\alpha_{f}=\frac{I_{f}(\beta +\varphi )}{v}$$
$$\alpha_{f}=\frac{I_{r}(\beta -\varphi )}{v}$$

It is possible to compute the lateral and yaw dynamics of the vehicle. The equation for the lateral dynamics is:$$\beta =\frac{\left(C_{f}+C_{r}\right)}{mv}-\frac{\left(I_{f}C_{f}-I_{r}C_{r}\right)}{mv}+\frac{C_{f}}{m}\delta$$

Similarly, the yaw rate updated equation is as follows:$$\dot{\varphi }=\frac{(I_{f}C_{f}-I_{r}C_{r})\beta }{I_{z}v}-\frac{{\left(I_{f}C_{f}+I_{r}C_{r}\right)}^{2}}{I_{z}v}+\frac{I_{f}C_{f}}{I_{z}}\delta$$

The vehicle lateral state space model is defined and stated as:$$\frac{d}{dx}\left[\begin{array}{c} y_{0}\\ \varphi \\ \beta \end{array}\right]=\left[\begin{array}{ccc} 0 & 1 & 0\\ 0 & -\frac{\left(C_{f}+C_{r}\right)}{mv} & 0\\ 0 & 0 & \frac{{\left(I_{f}C_{f}+I_{r}C_{r}\right)}^{2}}{I_{z}v} \end{array}\right]$$

Simulation controls require the vehicle dynamic plant model to find cross-track error.

### 3.4. Introduction to Steady-State Equations

In steering angle control system analysis and design, it is crucial to evaluate the whole system response and to develop controllers in such a way that a satisfying response is produced for all time instants *t*(0), where *t*(0) is the initial time. The system reaction is known to have two components: transient response and steady-state response, that is:(14)$$y\left(t\right)=y_{tr}\left(t\right)+y_{ss}(t)$$

The transient response is present for a short period of time and then disappears. If the system is stable, the transient response may potentially be recorded:$$\underset{t\to 0}{\lim}y_{tr}\left(t\right)=0$$

In addition, if the system is unstable, the transient response will grow extremely quickly (exponentially) in time, resulting in the system becoming completely unusable or even destroyed in most circumstances during the unstable transient response. It is critical in control systems that steady-state response values are as close to the desired (specified) ones as possible, so we must investigate the corresponding errors, which represent the difference between the actual and desired system outputs at steady state and examine the conditions under which these errors can be reduced or even eliminated.

The following set of nonlinear coupled differential equations of motion control of the vehicle are represented in the main body co-ordinate frame B: the steering angle is the input, while the mass centre is forward velocity, and the lateral velocity and yaw angle are the outputs in this equation. The steering angle issue can be investigated from other perspectives. The steering angle required to keep the vehicle in between different lanes will be calculated in this section. Figure 5 shows a rigid vehicle in planar motion, with the global (G) and body (B) frames fixed to the ground as well as to the vehicle’s mass centre, respectively.

A rigid vehicle is supposed to behave like a box on a horizontal surface (planar motion), with three degrees of freedom: *x* and *y* translation and rotation around the *z*-axis. In the body co-ordinate frame B, the Newton–Euler equations of motion for a planar rigid vehicle are:(15)$$v_{x}=\frac{F}{m}+rv_{y}$$

The car’s yaw rate is r = φ = z, and the front and rear wheels’ steer angles are the cot-average of the associated left and right wheels.

### 3.5. Speed Control for Sharp Curve Road

The longitudinal speed controller is built in this study using the derivation of the trajectory curvature $K_{tarjectory}$ and the vehicle curvature $K_{vehicle}$. While monitoring the reference trajectory, the autonomous vehicle calculates $K_{tarjectory}$ and $K_{vehicle}$ repeatedly. The trajectory and vehicle curvature may be computed iteratively as:$$K=\frac{∆\delta }{∆L}$$
where *K* and *L* are the trajectory’s curvature, central angle, and length. Based on the current curvature, the curvature function f ($K_{tarjectory}$, $K_{vehicle}$) value is utilised to calculate a suitable speed decrease for the autonomous vehicle while moving in a risky curve. We extract the new required velocity $V_{d}$ by subtracting the curvature function from the present velocity $V_{c}$, which prevents the vehicle from cutting corners while following the track.
$$V_{e}=K_{d}-V_{c}$$

The estimated velocity error $V_{e}$ is supplied into the MPC controller, which accurately adjusts the throttle pedal position to maintain the ideal speed. The MPC velocity controller equation is as follows:$$U_{v}=K_{p}V_{c}+K_{d}\frac{∆V_{e}}{∆t}$$

Furthermore, we can calculate the slip angle of the vehicle as well as the required traction force to maintain a constant forward speed.
(16)$$F_{x}=-\frac{mv_{x}}{r}$$
(17)$$\beta ={tan}^{-1}\frac{v_{y}}{v_{x}}$$

When the vehicle is turning at a steady-state condition on straight and clothoid-shaped roads, it is governed by the following equations. As illustrated in the figure, the ground has a global co-ordinate frame G, while the car mass centre C has a vehicle co-ordinate frame B. Z is considered to be parallel and the angle shows how B in G is oriented when φ is the angle of the heading of a vehicle.

When a vehicle is travelling in the first quadrant:$$\mathrm{Steering}\;\mathrm{angle}=\frac{a^{2}}{\pi S}cos(\frac{\pi S^{2}}{2b^{2}})\;\cos\varphi +\frac{a^{2}}{\pi S}\sin\left(\frac{\pi S^{2}}{2b^{2}}\right)sin\varphi$$

When a vehicle is travelling in the second quadrant on a clothoid road:$$\mathrm{Steering}\;\mathrm{angle}=\frac{a^{2}}{\pi S}cos(\frac{\pi S^{2}}{2b^{2}})\;\cos\varphi -\frac{a^{2}}{\pi S}\sin\left(\frac{\pi S^{2}}{2b^{2}}\right)sin\varphi$$

Based on the above equation, we can define the curvature response and steady-state response:$$S_{k}=\frac{k}{\delta }=\frac{1}{R_{\delta }}$$

Yaw angle

The yaw angle is the angle between the longitudinal axis of the vehicle and an axis parallel to the surface of the Earth in an Earth-fixed co-ordinate system.
(18)$$S_{f}=\frac{r}{\delta }=\frac{k}{\delta }v=S_{k}v$$

Centripetal acceleration

A body travelling in a circular direction will experience centripetal acceleration, which is the acceleration of the body. Given that velocity is a vector quantity (that is, it has both a magnitude, which is the speed, and a direction), when a vehicle travels in a circular path, the direction of the body continually changes, causing the body’s speed to vary, resulting in the body experiencing an acceleration.
(19)$$a=\frac{v}{r}∆S$$

Lateral velocity
V= vcosφ
(20)$$v=V_{x}+\int_{t_{0}}^{t}a_{r}dt$$

Surface sideslip angle
$$\beta -\frac{V}{R}$$

As previously stated, this route will be utilised and is made up of two separate clothoid and straight roads. The clothoid is used as a sample road for determining a vehicle’s kinematic steering angle as an example of how the lane detection and tracking algorithm can maintain a vehicle kinematically on the road. Figure 6 shows the desired path of motion by minimising cross-track error. 

The parametric equation of the road, which is moving in the *X* direction and starts from the origin, is as follows. The parameter t is not constant and it varies in all equations of motion.
(21)$$X\left(t\right)=\int_{0}^{t}\cos(\frac{\pi }{2}\times v^{2})dv\;\;\;\;\;\;0\leq t\leq 1$$

### 3.6. Rotation Centre

The Laplace transform is applied to the vehicle equations of motion for the steering wheel angle and the rear wheel steering angle and, when the yaw rate response to the steering angles is solved, the following results are obtained:(22)$$Y(s)=\frac{\frac{1}{n}G_{\delta }\left(0\right)(1+\frac{1}{2K}S^{2})}{1+\frac{2}{\omega_{n}}S+\frac{1}{w_{n}^{2}}}$$

The yaw rate is calculated using the first-order lag yaw rate. The impact of altering the steering angle:(23)$$\delta_{G}\left(t\right)=F_{SG}\delta_{F}(t)$$

We can calculate the steady-state location of the centre of the vehicle using the steady-state response:(24)$$\delta -\frac{l}{R_{1}}=\alpha_{R}-\alpha_{F}$$
and:(25)$$\delta -\frac{l}{R_{2}}=\alpha_{R}+\alpha_{F}$$
where $R_{1}$ and $R_{2}$:$$R_{1}=\frac{1}{cos2\alpha }\left(xcos\varphi -ysin\varphi \right)$$
$$R_{2}=\frac{1}{cos2\alpha }(ycos\varphi -xsin\varphi )$$

This results in a greater turning radius for the front wheels, which is normally in order to track the tracking at a point on a tangent to the turn circle of the rear wheels. It will be shown that the steady-state response equation is sufficient for predicting the transition behaviour of a vehicle in a steady state. It will also be considered whether a step steer angle adjustment and a lane change steering input should be used. 

### 3.7. Change in Steering Angle

A realistic step change of the steering angle with fluctuating speed will be expressed by:(26)$$\delta =\delta_{0}(H\left(t-t_{0}\right)+{sin}^{2}(\frac{t}{2t_{0}})\;H\;(t_{0}-t))$$
$$H\;(t-t_{0})=\left\{\begin{array}{c} 0\\ 1 \end{array}\right.$$
where:$$t_{0}=response\;time$$
and:δ=constantSteerangle

To determine the reaction of the vehicle for a given value:δ=20 rad
$$t_{0}=30\;s$$
v=60 m/s

To determine the solution of the equation of motion:$$r=H\;(t-t_{0})e^{4.963}(0.0493sin3.7688\left(t-1\right))+0.2924\cos(3.7688\left(t-1\right))-H(t-t_{0})\;0.0588\;\sin\;(\pi t)+\;0.0870\;\sin\;(\pi t)+0.2439+e^{-0.493t}(0.2925\cos\left(3.7688t\right))+0.0493\sin(3.7888t)+0.1223\;\cos\;(\pi t)+0.2440$$
$$v=H\;(t-t_{0})e^{4.9622\left(1-t\right)}\left(0.0492\right)\sin3.7688\left(t-1\right)+0.2924\cos(3.7688t)+0.1223\cos\left(\pi t\right)+\sin\left(\pi t\right)-0.415$$
$$R=\frac{1}{k}$$

These are dynamic variables, which are calculated by solving the equation of motion where in 0-1 and l-1 are used. As can be observed in Figure 7, the actual steer angle for the left and right front wheels is not the same but is somewhat smaller for the left wheel and slightly bigger for the right wheel, respectively. In reality, this is accomplished by the use of a steering link mechanism but, if $s_{l}$ and $s_{d}$ are both small, the difference between the two wheels’ steer angles may be assumed to be the same, and the left and right wheels can be considered to have the same steer angle.

When the equations for the straight and clothoid roads are given, then the point that the car should turn around can be evaluated. The exact location of the curvature centre and adjusted angle of the steering of the vehicle to coincide with the vehicle’s turning centre can be used to calculate the kinematic motion of the vehicle.
$$\dot{x}=v_{car}cos\theta =u_{1}cos\theta$$

We know the equation of the straight and clothoid road and the values of the turning centre for the first and second quadrant. The clothoid is a curve in which the combination of the radius of curvature by the arch length is constant at every point along the curve. In addition to making it an excellent transition curve, it also provides for the straightforward calculation of the arc length parameterisation of the curve.
(27)$$\frac{v_{car}}{cos\theta }sin\emptyset =v_{car}tan\theta =v_{tangent}$$

For the steering angle for the front wheel in the first and second quadrant of the straight and clothoid road, the ratio of front-to-rear stiffness is expressed as:(28)$$C_{r/f}=\frac{C_{R}}{C_{F}}$$
(29)$$\delta -\frac{l}{R}=\frac{T}{lR}\left[1+C_{r/f}\right]+a_{y}K$$

It has features similar to the vehicle response to the front wheel steer angle. The vehicle’s response to the steering wheel angle is characterised by the following features. When considering the circular motion of the vehicle at higher speeds, the centrifugal force becomes more relevant. In order to counteract this centrifugal force, the cornering forces at the front and rear wheels must be applied, resulting in the production of sideslip angles. 

### 3.8. Effect of Acceleration

The impact of acceleration (varying forward velocity) on the steady-state and transient reaction of the turning centre and the motion of the vehicle is studied in this section. A comparison of the two stated vehicle responses is used to demonstrate that there is a small difference between the steady state and centre of rotation of the vehicle. The dynamics of a car with a fixed steering angle and changing forward velocity will be studied and reported. It has been shown that, by using a steady-state response, it is feasible to predict the vehicle’s dynamics within acceptable engineering applications. We will calculate the dynamic rotation centre of the vehicle and compare it to steady-state data. The outcome is essential in developing a lane detection and tracking algorithm for self-driving cars. The reaction of an understeer passenger vehicle travelling with a constant steering angle equal to the equation and a variable forward velocity that varies with time is calculated in the following equation. Solving an equation of motion will determine the vehicle’s transition behaviour, which will be identified using steady-state responses. It has been shown that steady-state response equations are adequate for predicting the car’s transition behaviour.
$$\delta \left(t\right)=0.1\;rad\approx 5.37\;deg$$

The forward velocity of the vehicle is directly proportional to time according to the following function:$$v_{x}=\frac{20}{t_{0}}tH\left(t_{0}-t\right)+10H\left(t-t_{0}\right)\;m/s$$
$$H(t-t_{0})=\left\{\begin{array}{c} 0\;t\leq t_{0}\\ 1\;t\geq t_{0} \end{array}\right.$$

The sideslip ratio is as follows:$$=1-\frac{(R+t)\delta }{\frac{r}{R}}\times 100$$
$$=\frac{t}{T}\times 100$$

An illustration of a clothoid is a curve in which the product of the radius of the curvature by the arch length is constant at every point along the curve. In addition to making it an excellent transition curve, it also provides for the straightforward calculation of the arc length parameterisation of the curve.
(30)cαL=F
(31)$$\alpha =\frac{1}{CL}\frac{t^{2}}{2R}$$

It follows mathematically that a slip angle, denoted by the letter f, is essential to counteract the understeer contribution given by the solid back axle. Clearly, significantly more effort will be required to enable the vehicle to turn around the bend. The following circumstances, which are adequate for a formula student car on a 20 m skid pan, may be used to make an estimate.

### 3.9. Longitudinal Response

Roll steer is defined as the angular displacement of the wheel caused by the roll of the vehicle. In contrast to a negative roll steer, which operates in the opposite direction of the real steer angle, a positive roll steer acts in the same direction as the actual steer angle. The geometry and relationship of a steady-state cornering vehicle with roll steer is seen in Figure 7. With the exception of roll steer, the geometrical relationship of steady-state cornering is provided by following Equation. When using roll steer, the equation is:(32)$$\delta =\frac{L}{P}+\delta_{r}-\delta_{f}$$

Roll steer, in addition to the steady-state steer angle, may be used to analyse vehicle steer characteristics in the connection between the steady-state steer angle and the lateral acceleration (*y*) when the roll steer is taken into account.

Sideslip angle of a vehicle for a clothoid road:$$S=\frac{y_{1}-0.5\times 10^{8}th\left(-t+20\right)-0.1\times 10^{10}h(t-20)}{y_{3}+0.25\times 10^{10}h\left(-t+20\right)+0.5\times 10^{9}h(t-20)}$$
where:$$y_{2}=\frac{0.9375\times 10^{10}}{th\left(-t+10\right)+20h(t-10)}$$
and:$$y_{3}=\frac{0.625\times 10^{11}}{th\left(-t+10\right)+20h(t-10)}$$

Yaw rate:$$Y=\frac{0.625\times 10^{10}}{y_{3}+0.25\times 10^{8}th\left(-t+10\right)+0.5\times 10^{9}h(t-10)}$$

Lateral velocity:$$L_{v}=\frac{\left(y_{3}-0.5\times 10^{8}\right)th\left(-t+10\right)-0.1\times 10^{10}h\left(t-10\right)th\left(-t+10\right)+10(t-10)}{y_{2}+0.25\times 10^{8}th\left(-t+10\right)+0.5\times 10^{9}H(t-10)}$$

We can calculate the steady-state location of the rotation of the centre of a vehicle in the vehicle body co-ordinate frame using the steady-state response $S_{k}=\frac{1}{R}$ AND S=L
(33)$${CO}_{steady}=-K_{x}(k)$$
and:(34)$${CO}_{steady}=K_{x}(k)+R\;\sin\alpha$$

The dynamic rotation of the understeer vehicle travels away and forward as the forward velocity of the vehicle increases at a constant steer angle. The rate of deployment of the rotation centre is directly proportional to the rotational speed of the vehicle, so it increases with displacement. The location of the centre of rotation in relation to the vehicle’s body frame varies with speed. At the critical speed, we have ∝=0 and the dynamic centre of rotation is on the *y*-axis. At the start of movement, the global frame G is fixed on the ground and *B* corresponds with G. The *B* travels with the vehicle, yet the *Z*-axis remains parallel at all times. As a result, the vehicle velocity vector in the global frame is:(35)$$\left[\begin{array}{c} F_{x}\\ F_{y} \end{array}\right]=\left[\begin{array}{cc} cos\delta & -sin\delta \\ sin\delta & cos\delta \end{array}\right]+\left[\begin{array}{c} F_{x}\\ F_{Y} \end{array}\right]$$

The body frame is given by velocity vector:(36)$$B_{v}=\left[\begin{array}{c} v_{x}\\ v_{y} \end{array}\right]$$

Therefore, the global co-ordinates of the mass centre of the vehicle would be:(37)$$X=(R_{N}+h)\left(\emptyset -\emptyset_{0}\right)+∆X$$
and:(38)$$Y=\left(R_{N}+h\right)\left(\emptyset +\emptyset_{0}\right)+∆Y$$

When the steer angle is constant, the vehicle reaches a maximum speed of 70 m/s or above and the vehicle will ultimately turn in on the clothoid route; the vehicle’s steady-state centre is located.
(39)$${CO}_{1steady}=-Rsin\propto \;\;CO_{1steady}=-\frac{1}{k}sin\propto$$
and:(40)$${CO}_{2steady}=Rcos\alpha \;\;CO_{2steady}=\frac{1}{R}cos\alpha$$

The global co-ordinates of the steady-state rotation centre are:(41)$$R_{N}=\frac{R_{major}(1-e{)}^{2}}{(1-e^{2}\emptyset {)}^{3/2}}$$

### 3.10. Look-Ahead Distance Effect

The effect of changing ks can be seen directly (the effect of the look-ahead distance examined at the end). A significantly larger error is not caused by the smaller control parameter, as was the case with the ks change. When the parameter is larger than the optimum, a larger error occurs. There is a resemblance between the two, with a larger value resulting in cutting corners and a smaller value resulting in a slower steering response.

The one parameter used is look-ahead distance. The impact of adjusting the look-ahead distance must be weighed against two problems:The vehicle is far from the path and this must be rectified.Path maintenance, i.e., the vehicle is on the path and wants to remain there.

If the controller has a small look-ahead distance parameter, the heading error affects the steering reaction more than the predicted distance error. Due to the short look-ahead distance, these errors start to increase at the same moment. 

The result is overshoot.

The yef response is virtually identical but occurs later; thus, reducing this distance parameter causes the vehicle to respond later to path changes, increasing the heading errors (Figure 8).

If the controller has larger look-ahead distance parameters, the vehicle begins steering before the real turn has been reached. When the vehicle is turned slightly to the right, the look-ahead distance crosses the path, resetting the computations such that the error distance yef is positive. This generates an oscillating response that decreases after a period of time, only to return at the end of the corner.

Another observation is that the acceleration values of the controller are higher than the simulation in both conditions (smaller look-ahead distance and larger look-ahead distance). Once the value of ks was changed, they both shrank. This is explained by evaluating the steering angle in Figure 9.

We learned some of the limitations of our path tracking method over the last year from the literature. The two main problems are connected to dynamics. The method assumes optimal responses to desired curvatures, since it does not simulate the vehicle or its actuators. This causes two problems:A dramatic change in curvature might cause the vehicle to rear.The vehicle’s path will not be stopped as soon as expected due to a first-order lag in steering.

Table 1 presents the relationship between lateral acceleration and its consequences for the passenger when travelling at different speeds.

The algorithm used to calculate the front wheel steering angle utilizing MPC preview capabilities is depicted in Figure 10. The vehicle controller is made up of three components: the reference generation, the MPC optimiser, and the vehicle model. The reference generation module estimates the vehicle state and precomputes the reference yaw rate. The anticipated vehicle condition at the following sample time will change depending on the lateral acceleration determined by the controller at every sampling time of every prediction horizon. The reference yaw rate is then derived by the repeated process of obtaining the ideal preview lateral acceleration and foretelling vehicle states. The changing state variables at the next sampling period will then result in a new optimal preview lateral acceleration. In general, in MPC, the state estimation is finished all at once but, when the preview capability is used, the lateral vehicle speed and acceleration will change at the next sample time. As a result, the rolling computation is essential for lowering the cumulative error and obtaining more precise reference yaw rates (step1). The reference is made up of two modules: the reference yaw rate and the reference lateral displacement. The planned trajectory in the preview distance superimposed on the predictive horizon is used to produce the reference yaw rate, whereas the reference lateral displacement is derived from the reference route in the predictive horizon. When compared to the general MPC, our previously developed learning-based lane detection approach [24] is applied in step 2 to lengthen the effective reference path without adding to the workload. In order to follow the reference path, the MPC optimiser with input and output constraints calculates the steering wheel angle, and the solution of the MPC-PC joint control method with constraints is converted into a quadratic programming problem with constraints (steps 3 and 4).

## 4. Experimental Results and Discussion

### 4.1. Learning Based Lane Detection Simulation Model

This research article mainly concentrates on mathematical model development from steady-state dynamic motion equations to find key parameters of the learning-based lane detection algorithm, such as yaw angle, sideslip, and steering angle. We have applied and tested this algorithm to develop a simulation model for lane detection for straight roads in our previous study [24], which has not been tested for clothoid-form roads. Therefore, we apply the same algorithm for simulation experiments for the clothoid-formed roads in the present study. More details on the evaluation of the algorithm and the procedure of the simulation model can be found in [24]. The image processing and lane detection algorithm developed provides the inputs to the MPC controller. The middle line of the car is the centreline, which is used to compute the offset of the car position from this centreline and the yaw angle. This information is used by the MPC controller that tries to keep the car on the desired path on unstructured roads. In addition, the front view of the car is captured with a camera that is mounted on top of the car. Offset distance from the region of interest (ROI) and bird-eye view can be determined automatically and adaptively in every frame. Likewise, offset distance from the centreline is calculated and the yaw angle is adjusted so that algorithm detects the lane; so, the self-driving car can be controlled to stay within a lane on unstructured roads. These major steps involved are summarised and shown in Figure 11. 

### 4.2. Experiments

Experiments were conducted to ensure the precision and strength of the proposed method. We analyse the impact of the parameters and compare the results from testing the proposed networks in a wide range of climates and atmospheric conditions. In this research, a lane identification algorithm was tested in a simulated driving environment using videos of actual roads. Real-time footage captured by a car’s camera was used in the experiment, and lane lines were identified in a variety of challenging scenarios (e.g., highways and structured and unstructured roads).

### 4.3. Datasets

Based on the TuSimple lane dataset (Global autonomous driving technology company, San Diego, CA, USA) [25], BDD110K [26], KITTI [27], and our own lane dataset, we created a set of data. In total, there are 3626 image sequences in the TuSimple lane dataset. In these pictures, your forehead replaces the highway. Each sequence comprises 20 consecutive, one-second-long frames. The lane ground truth labels are applied to the 20th image in each sequence. Every 14th image in each sequence was labelled to expand the dataset (randomly selected; Table 2). We added over 1600 image sequences of rural roads to our own lane dataset. Since then, the lane dataset has grown substantially richer in variety. In addition, testing datasets were created using the interpolation technique [28]. This technique was used to perform a dynamic analysis of the lane recognition system in the simulation test experiment. Interpolation was originally developed as a method for testing software and hardware prototypes.

In order to train the proposed network and correctly identify lanes in the last frame, we used a sample of 1600 continuous images and the ground truth of the last frame as input. The training set was built from the ground truth label on the 18th and 26th frames (which were obtained in the previous step). Meanwhile, we sampled the input images at three different strides, i.e., at an interval of one, two, and three frames, to fully adapt the proposed network for lane detection at different driving speeds. Then, as shown in Table 2, three distinct sampling strategies can be used for each ground truth label. In data augmentation, operations such as rotation, flip, and crop are applied to generate a total of 1600 sequences, with 1600 labelled images used for training. The input was randomly transformed into new lighting conditions, expanding the dataset’s usefulness. Ten continuous images were sampled for testing, with the goal of lane detection in the last frame and comparison to the last frame’s ground truth. We developed a pair of totally separate test datasets.

There were two sets of tests. The first test set (TuSimple, BDD100K, and KITTI) was designed for typical testing. The second set of testing data comprised realistic examples taken from a variety of real-world scenarios in order to gauge robustness.

We also tested our method (proposed algorithm) with image sequences where the driving environment changed dramatically, namely, a car coming into and out of structured and unstructured roads on clothoid. The result shows the robustness of our method. We compared the proposed methods to other methods reported in the TuSimple lane detection competition to further confirm the excellent performance of the proposed methods. The TuSimple, BDD100K dataset served as the basis for our selection of training data. In contrast to the pixel-level testing standard we used previously, in this case, we adhered to the TuSimple, BDD100K testing standard, sparsely sampling the prediction points. Since crop and resize were used during the preprocessing phase of creating our dataset, we first mapped them to their original image size. Figure 11 shows that our FN and FP are very competitive, with the best results, and have the highest accuracy of all methods tested. The results from the TuSimple competition show that the proposed framework performs well when compared to state-of-the-art methods. We also used our dataset (interpolation) to train and test our networks and Pan’s approach, both with and without additional training data. These methods achieve marginally lower accuracy, higher FP, and lower FN when no supplemental data are used.

### 4.4. Implementation of Details

The resolution of the images used for lane detection in the experiments was 240 × 560. Windows 10 64-bit, MATLAB (2022a), and the Driving Scenario Designer program were all part of the simulation test environment. The system had a 3.20 GHz Intel Core i5-6000 CPU, 16 GB of RAM, and a two-terabyte hard drive. The model predictive control (MPC) was built using the MATLAB Model Predictive Control ToolboxTM, which includes the necessary functions, an app, and Simulink^®^ blocks. Different testing conditions, such as wet, cloudy, and sunny scenes, as well as a clothoid road, were used to verify the relevancy of the low-resolution images and the effectiveness of the proposed detection method.

### 4.5. Robustness of Lane Detection and Tracking Algorithm

Even though the proposed lane detection model did well on the previous test dataset, we still needed to test how well it works in real life. This is because even a small mistake can make it more likely that a car accident will occur. A good lane detection model should be able to handle a wide range of driving situations, from everyday scenarios, such as driving on a city street or highway, to more difficult ones, such as driving on a rural road with poor lighting and vehicles in the way on clothoid roads.

A new dataset consisting of simulated and actual driving scenarios was used to test the system’s reliability. As explained in the dataset section, test set #2 consisted of 1600 images with lanes in highway scenes (structured and unstructured roads). This dataset was recorded by a data recorder (monocular camera mounted on the top of the vehicle) at different heights, inside and outside the front windscreen, in different weather conditions and generated ground truth using an interpolation approach (linear and cubic spine interpolation). It is a large and difficult test set, with some lanes that are so hard to see that even humans fail to identify them. Figure 12 shows the lane detection model developed to evaluate the performance of the proposed algorithm. Table 3 shows the accuracy of the proposed algorithm at different times. 

Figure 12 shows how effectively the suggested method worked in different settings. With a mean processing time (per frame) of 20 ms, the lane detection accuracy reached 99% (milliseconds). Overall, the accuracy varies from 98.7% to 99%, with detection times ranging from 20 ms to 22 ms. In comparison to lane detection in the driving video sequences, the mean lane detection rate was marginally lower and the mean time interval (per frame) was much longer. However, in the BDD100K, TuSimple, and KITTI datasets, the suggested approach still outperformed the competition while maintaining adequate accuracy and adaptability. An intersection detection matrix was used to evaluate the performance of the algorithm.

Figure 12 and Figure 13a,c show some of the proposed algorithm’s results before any postprocessing. Lanes in difficult situations are identified perfectly, even when the lanes are hidden by cars, shadows, or dirt and when the lighting and road conditions are different. In some extreme situations, such as when all of the lanes are covered by cars and shadows or when the lanes are slanted because of seams in the road structure, etc., the proposed models can still identify them. The proposed models also work well with different camera angles and positions. As shown in Table 4, test 3 is more accurate than the others in all scenes by a large margin and obtains the highest F1 values in most scenes, which shows that the proposed models are superior.

We also tested our methods with image sequences that show considerable changes in the driving environment, such as when a car goes into and out of the shade. Figure 14 shows how well our method works.

Table 4 shows that the accuracy and F1 measure increase when more consecutive images are used as input with the same sampling stride. The benefits of the proposed network design using multiple consecutive images as input are demonstrated. The methods that take in more than one image are much better than the methods that only take in one image. As the stride length becomes longer, the performance tends to stay the same. For example, going from four frames to five frames does not improve the performance as much as going from two frames to three frames. This could be because information from frames farther back is less useful for predicting and identifying lanes than information from frames closer to the present. Then, we examined how the other parameter, which is the sampling step between two consecutive input images, affects the outcome. From Table 4, we can see that when the number of frames stays the same, the proposed models perform very similarly at different sampling rates. In fact, the effect of sampling stride can only be seen in the results down to the fifth decimal place, meaning that the sampling stride does not seem to have much of an effect.

The developed lane detection approach was compared with other algorithms published in the current literature to demonstrate its superiority. In this study, the proposed algorithm was contrasted with learning-based methods and traditional detection techniques (Table 5). Similarly, in [29,30,31,32,33], the suggested algorithm was used to analyse all pertinent lane recognition tests on the primary and secondary dataset on various road geometries. In addition to providing a thorough performance comparison for accuracy measures, the results reveal that the proposed algorithm, which is based on a learning-based approach, performs better than more conventional methods, demonstrating the robustness of the proposed system in this research work.

### 4.6. Visual Examination

A high-quality neural network for semantic segmentation should be able to accurately divide an input image into discrete regions, both at the coarse and fine levels of detail. The model is required to accurately predict the total number of lanes in the images at the coarse level. Lane detection processing should take care to avoid two specific types of detection errors. Both missing detection and excessive detection result in incorrect predictions of background objects as lanes, with the former occurring more frequently. These two types of detection errors will have a negative and far-reaching impact on ADAS judgement because they will lead to discrepancies in the predicted and actual number of lanes.

### 4.7. Running Time

Due to the proposed models’ use of time-series data, which requires processing a series of images as input, the proposed models may be more resource-intensive to run. When compared to other lane detection models that only process a single image, which uses an image segmentation block, such as SegNet and U-Net [34,35], the proposed algorithm can still reduce the processing time by 20–22 ms when SegNet and U-Net is not applied to all 1600 frames. If the proposed methods are implemented online, where the encoder network only needs to process the current frame because the previous frames have already been tested, the running time can be significantly reduced. Due to the fact that GPUs can run the ConvLSTM block in parallel, the ConvLSTM model is one of the most interesting deep-learning blocks that is used to predict next-frame video or image, the per-frame processing time is only about 20–22 ms, and this is almost identical for models that only use a single image as input.

### 4.8. Robustness

While the proposed lane detection model has shown promising results on previous test datasets, its robustness still needs to be verified. This is because any misidentification, no matter how slight, can raise the probability of an accident. To be effective, a lane detection model must be adaptable to a wide range of driving conditions, from the typical urban road and highway to the more difficult rural roads, poor illumination, and vehicle occlusion on both structured and unstructured roads. For the purpose of testing robustness, we employed a newly created dataset based on interpolation and secondary data (BDD100K dataset) that contained numerous actual driving scenes. The data in this set were recorded by a device mounted on the dashboard at varying heights, both inside and outside the front windshield, and in a variety of climatic conditions. Detecting some lanes is difficult enough for human eyes, making this a comprehensive and difficult test dataset.

## 5. Conclusions

This study proposes a novel steady-state dynamic control for robust lane recognition in a driving situation for clothoid-form roads. Two different situations are offered and examined to analyse the features of an automobile on a clothoid road: constant steering angle and variable longitudinal velocity, and variable steering angle and variable longitudinal velocity. The proposed network architecture is built on a framework based on learning that receives several continuous frames as input and predicts the lane of the current frame using semantic segmentation.

Simulation tests for the lane detection approach were performed using a road driving video in Melbourne, Australia, as well as the Berkeley DeepDrive Industrial Consortium’s BDD100K dataset, TuSimple, and KITTI dataset. With our suggested approach, the mean detection accuracy varies from 97% to 99% and the detection time ranges from 20 to 22 ms under various driving circumstances. In terms of efficiency and overall performance in real time, as well as detection efficiency and anti-interference abilities, the suggested lane detection algorithm was found to be superior to traditional techniques and learning-based approaches. Both the accuracy and mean time interval were significantly improved. When compared to existing controllers, the performance of the suggested technique demonstrates a considerable reduction in tracking errors. The suggested technique contributes by estimating the kind of future sharp curves and computing the proper speed and steering angle for each curve to drive the autonomous vehicle, which is the desired aim of any autonomous vehicle in real-world driving situations. When the route curvatures are normal, the vehicle maintains a steady speed by appropriately managing the steering angle. If the impending curves are sharp, the car slows down before approaching them and achieves the correct speed and steering angle to avoid lateral mistakes.

In terms of lane identification accuracy and algorithm time reductions, the suggested lane detection algorithm displayed considerable gains. In addition to playing an important role in terms of driving assistance, our algorithm significantly enhanced the driving safety of autonomous vehicles in real-world driving conditions and effectively met the real-time goals of self-driving cars. Furthermore, the lane recognition algorithm’s inclusiveness and accuracy might be further optimised and improved to boost the method’s overall performance. First and foremost, the whole model should be tested using a simulator that simulates real-world road settings utilising input photographs and delivering feedback from the vehicle model. The suggested model outperformed existing models, with higher precision, recall, and accuracy values. Furthermore, the proposed model was tested on a dataset with very difficult driving circumstances to demonstrate its robustness. The results demonstrate that the proposed models can recognise lanes in a range of situations while avoiding false positives. Longer sequences of inputs were demonstrated to improve parameter analysis performance, confirming the idea that many frames are more advantageous than a single image for lane identification. We want to enhance the lane detection system in the future by including lane fitting into the proposed framework. As a consequence, the identified lanes will be smoother and more consistent. 

## Figures and Tables

**Figure 1 sensors-23-04085-f001:**
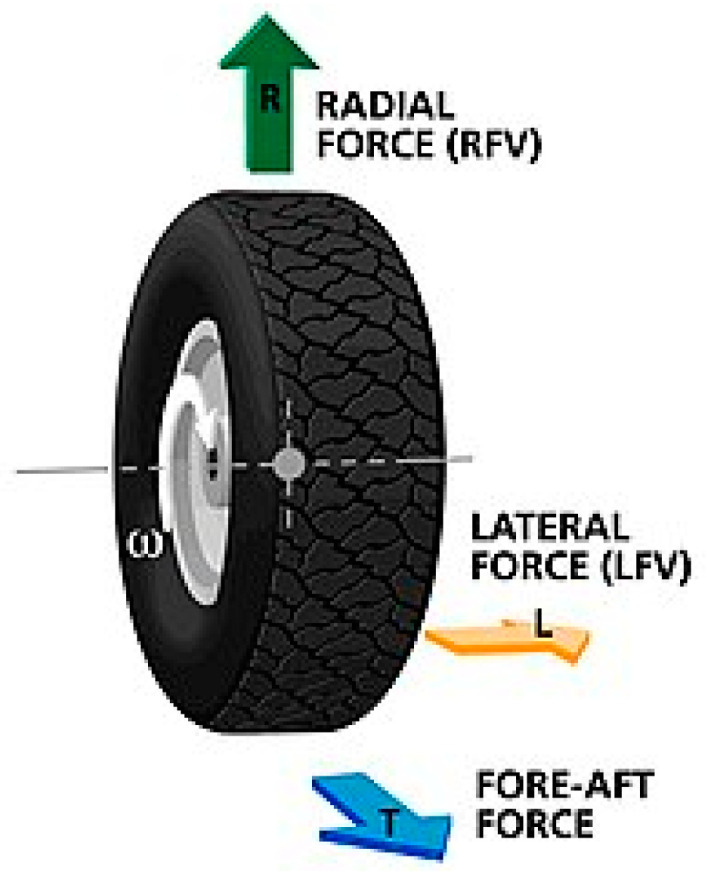
Tyre model.

**Figure 2 sensors-23-04085-f002:**
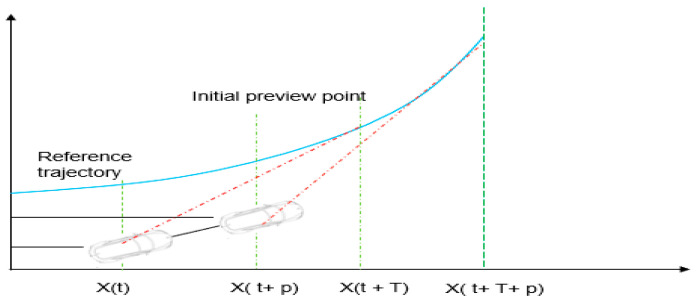
Prediction horizon combing MPC and PC.

**Figure 3 sensors-23-04085-f003:**
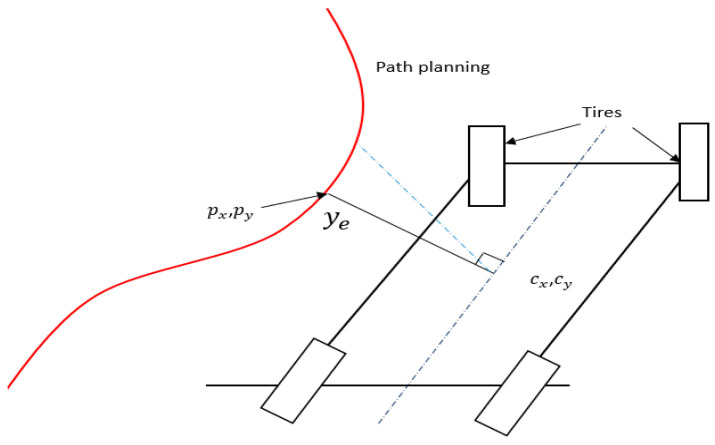
Four-wheeler model and perpendicular distance to the trajectory.

**Figure 4 sensors-23-04085-f004:**
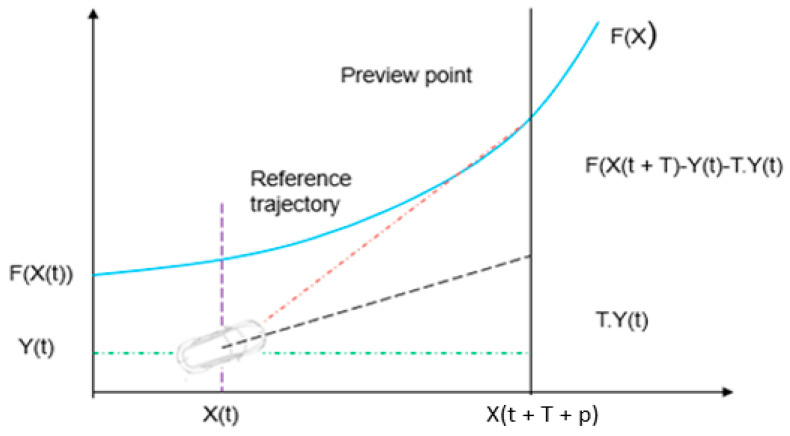
Preview capability of MPC.

**Figure 5 sensors-23-04085-f005:**
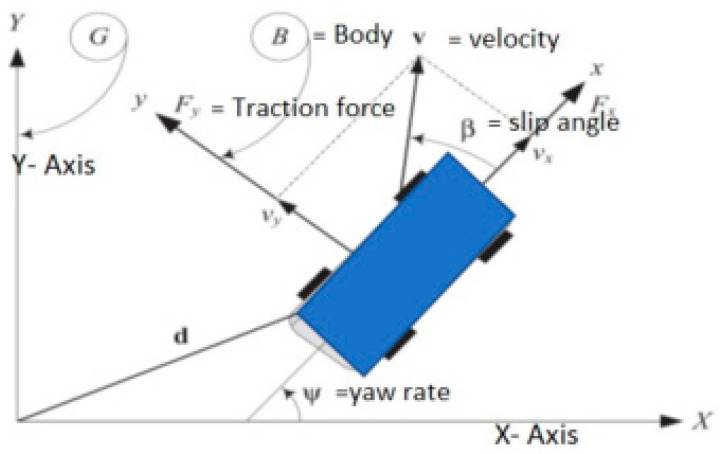
Forces acting on a vehicle when travelling on clothoid road. Adopted and modified from [23].

**Figure 6 sensors-23-04085-f006:**
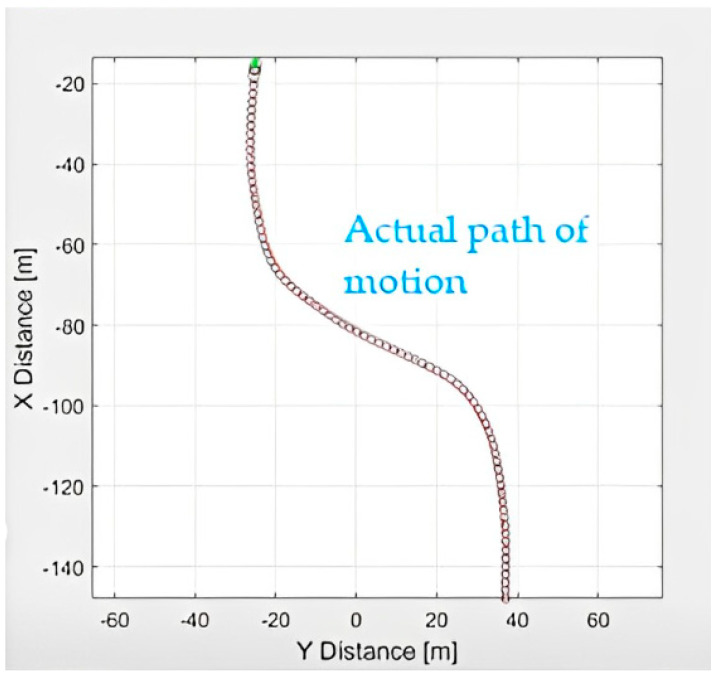
Desired path of motion.

**Figure 7 sensors-23-04085-f007:**
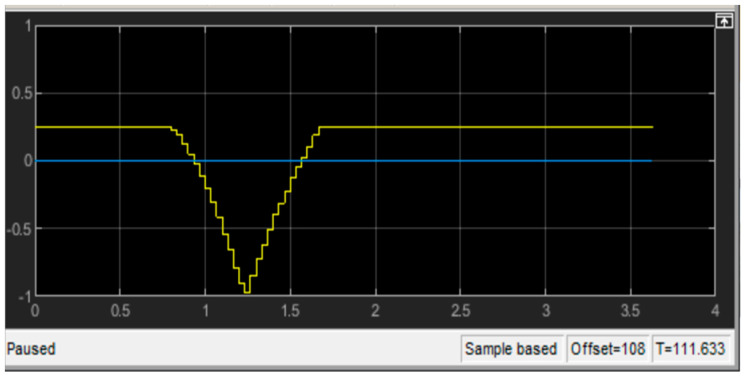
Exact location of the curvature.

**Figure 8 sensors-23-04085-f008:**
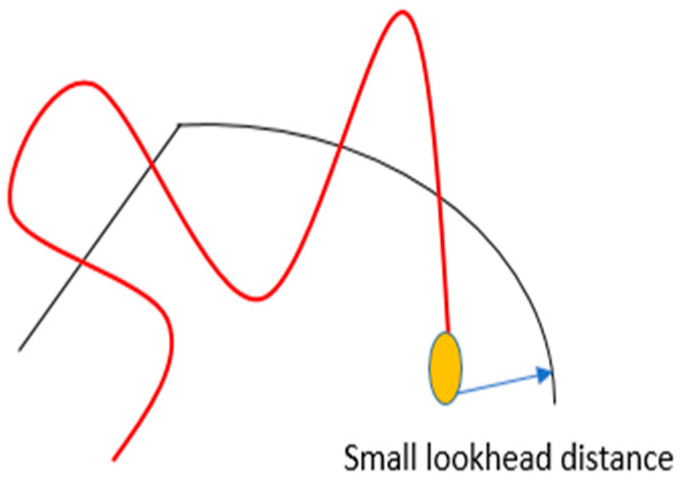
Small look-ahead distance.

**Figure 9 sensors-23-04085-f009:**
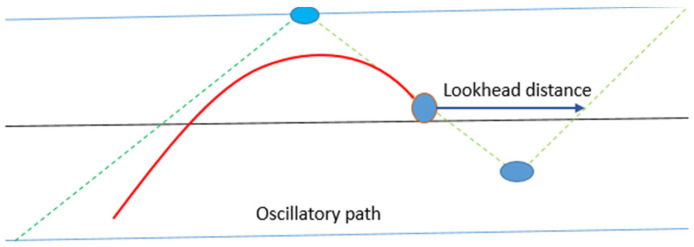
Larger look-ahead distance.

**Figure 10 sensors-23-04085-f010:**
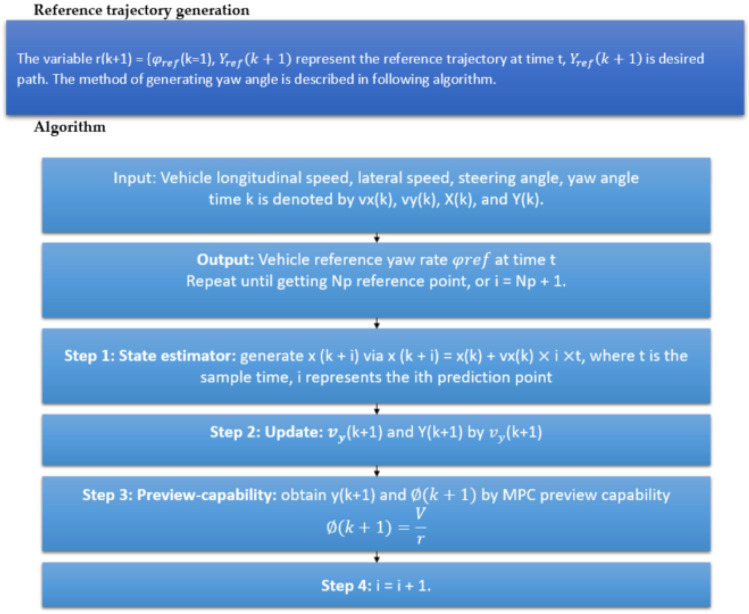
Steps involved in the proposed MPC-PC strategy incorporating a learning-based lane detection algorithm.

**Figure 11 sensors-23-04085-f011:**
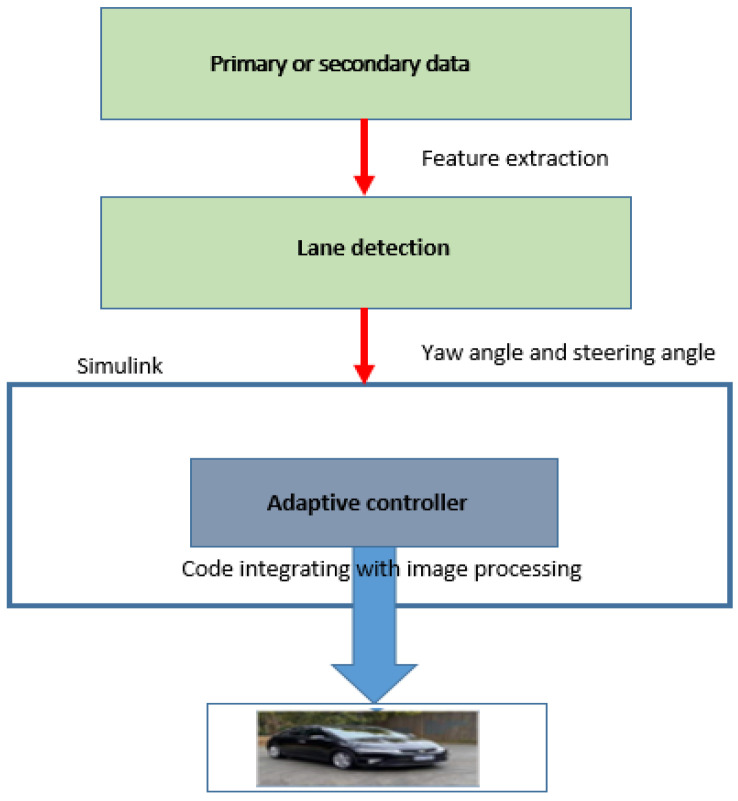
Working procedure of the lane detection algorithm.

**Figure 12 sensors-23-04085-f012:**
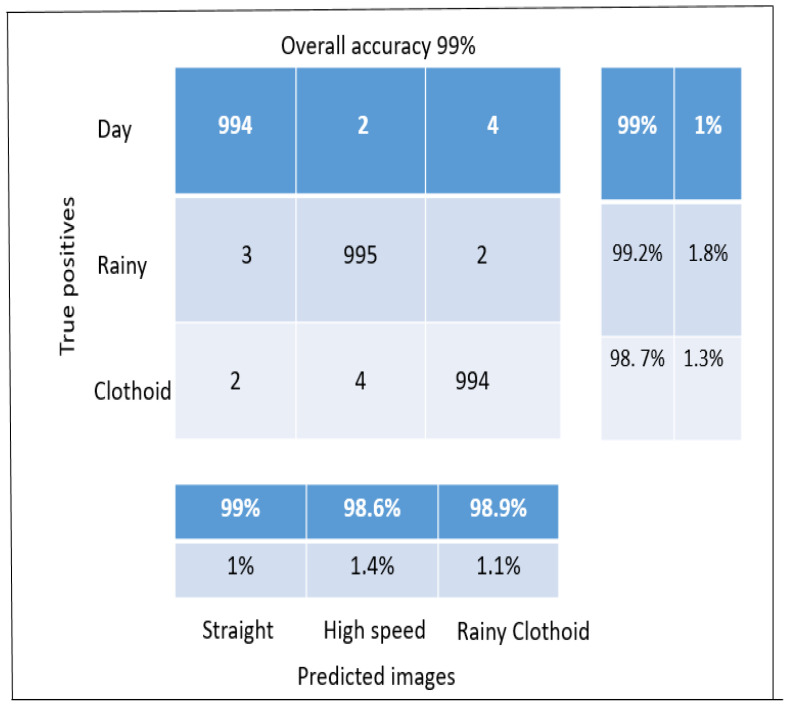
Performance of the proposed lane detection model.

**Figure 13 sensors-23-04085-f013:**
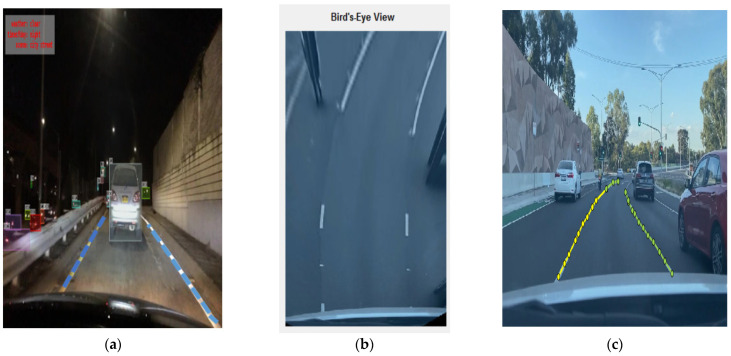
Lane detection using BDD100K secondary dataset (**a**), bird’s eye view for close view of lane (**b**), and primary dataset (**c**).

**Figure 14 sensors-23-04085-f014:**
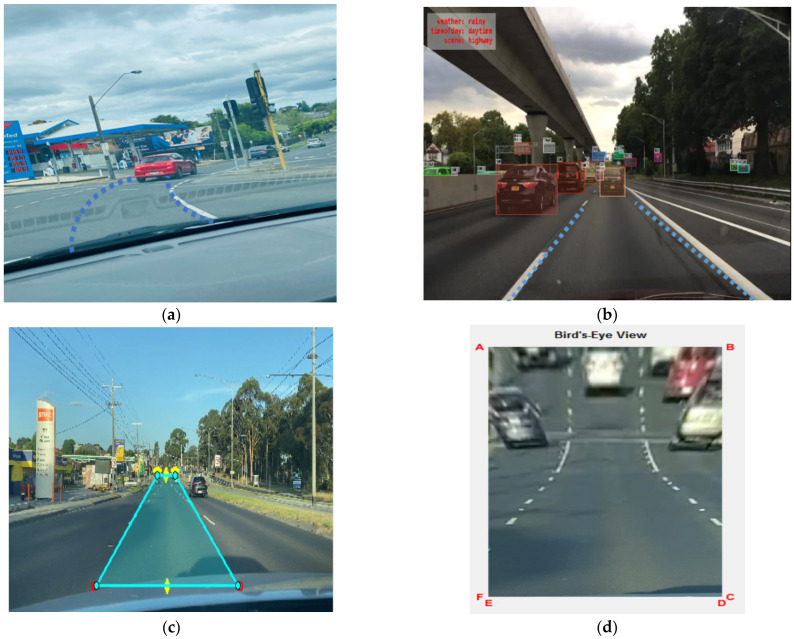
Lane detection and tracking on a structured road: primary data (**a**,**c**), BD100K secondary data (**b**), and bird’s eye view for close view of the lane (**d**).

**Table 1 sensors-23-04085-t001:** Consequences of lateral acceleration and its effects on passengers.

Lateral Acceleration	Consequences
0 $\leq a_{y}\leq$ 1.7	Comfort
$1.7\;\leq a_{y}\leq$ 3.7	Medium comfort
$3.8\;\leq a_{y}\leq$ 5.1	Discomfort
$5.1\;\leq a_{y}\leq$ above	Uncomfortable

**Table 2 sensors-23-04085-t002:** Training and testing datasets.

Type	Primary and Secondary Data	Labelled Images	Labelled Frame
Training set	BDD100K (Highway), KITTI, TuSimple (Our dataset) Interpolation approach	547 312	14th and 21st 18th and 26th
Testing set	Test 1 Test 2	1600 images	-

**Table 3 sensors-23-04085-t003:** Lane detection accuracy of the algorithm under different scenarios.

Video Sequence	No. of Positive Samples	No. of Negative Samples	No. of True Negatives	No. of True Positives	True Negative Rate	True Positive Rate	Accuracy
1. Day time	1504	34	05	03	1%	99%	99.7%
2. Night time	1145	56	07	05	1.4%	98.06%	97.9%
3. Rainy	1222	68	09	06	1.6%	98.07%	97.3%
4. (Day time)	1404	57	08	09	2.3%	97.07%	94.7%
5. (Night time)	1290	78	11	07	0.4%	99.06%	100%

**Table 4 sensors-23-04085-t004:** Performance evaluation of the algorithm.

Algorithm	Test Accuracy		Recall	F1-Measure	Running Time	Precision
	Clothoid	Highway Rural				
(Test#1)	97.35%	97.67%	0.985	0.892	0.0043	0.786
(Test#2)	98.46%	97.98%	0.974	0.895	0.0041	0.787
(Test#3)	98.37%	98.56%	0.948	0.897	0.0049	0.863

**Table 5 sensors-23-04085-t005:** Comparison of our results with existing literature.

Methods	Road Geometry	Accuracy Rate (Exiting Literature)	Accuracy Rate (This Study)
[29] Traditional method	Structured road	<97.00%	97.67%
[30] Spatial Ray Feature extractions	Straight road	94.40%	97.98%
[31] Hough transform	Structured road	95.70%	98.56%
[32] Fast Draw Resnet	Structured road	95.2%	99.7%
[33] ConvLSTM (Deep learning)	Unstructured road	97.3%	97.9%

## Data Availability

Not applicable.

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
