# Peer review of "Implementing Model Predictive Control and Steady-State Dynamics for Lane Detection for Automated Vehicles in a Variety of Occlusion in Clothoid-Form Roads"

_sensors, 2023, doi:10.3390/s23084085_

Round 1

Reviewer 1 Report

The manuscript presents a lane detection method based on Model Predictive Control and Steady-State dynamics. The paper is well-written and well-organized. The authors explain in detail the proposed method and use standard datasets to show the technique's effectiveness. However, the main drawback of the paper is the lack of an exhaustive comparison in terms of accuracy and execution between the proposed technique and the state-of-the-art methods. In my opinion, it is necessary to include a table with a proper comparison with other techniques.

The quality of the images, such as Figure 6, should be improved.

Author Response

Thank you for your feedback, Please find attached the response statement. 

Reviewer 2 Report

1. The author does not explain what the contribution of this article is. The author only mentions "This paper proposals a lane detection method using Model Predictive Control (MPC)" in the abstract, but the full text does not mention how this lane detection method actually does.

2. The author wrote in Introduction that "Section 3 introduces the proposed mathematical model for the lane detection algorithm that includes MPC preview capability strategy and steady state dynamics." However, how lane detection is accomplished is not mentioned in Section 3. I don't really understand how a lane detection model uses MPC. I think they are two separate steps and I hope the author can explain that.

3. The author seems to have introduced the algorithm in only a few sentences in Figure 10, but can't see how the author has modified the MPC algorithm?

4. It seem that the author does not  explain the metrics of MPC algorithm.

5. In Section 4.3, the author describes that all of his experiments are the result of lane detection, but no comparison with any existing algorithms is given.

6. In Section 4.5, the author proposed "When compared to models that only process a single image, such as SegNet and U-Net, the proposed algorithm can reduce the processing time by 20-22ms when applied to all 1600 frames." However, SegNet and U-Net are not lane detection models. If the authors have improved these two networks for lane detection, they should be described in the previous section.

Author Response

(The authors gave the same response as above.)

Round 2

Reviewer 1 Report

The authors have followed my suggestions.